# Design, Synthesis, and Antioxidant and Anti-Tyrosinase Activities of (*Z*)-5-Benzylidene-2-(naphthalen-1-ylamino)thiazol-4(5*H*)-one Analogs: In Vitro and In Vivo Insights

**DOI:** 10.3390/molecules30020289

**Published:** 2025-01-13

**Authors:** Hee Jin Jung, Hye Jin Kim, Hyeon Seo Park, Hye Soo Park, Jeongin Ko, Dahye Yoon, Yujin Park, Pusoon Chun, Hae Young Chung, Hyung Ryong Moon

**Affiliations:** 1Department of Manufacturing Pharmacy, College of Pharmacy and Research Institute for Drug Development, Pusan National University, Busan 46241, Republic of Korea; hjjung2046@pusan.ac.kr (H.J.J.); khj3358@pusan.ac.kr (H.J.K.); gustj6956@pusan.ac.kr (H.S.P.); hyesoo0713@pusan.ac.kr (H.S.P.); jungin8633@pusan.ac.kr (J.K.); dahae0528@pusan.ac.kr (D.Y.); 2Department of Medicinal Chemistry, New Drug Development Center, Daegu-Gyeongbuk Medical Innovation Foundation, Daegu 41061, Republic of Korea; pyj1016@kmedihub.re.kr; 3College of Pharmacy and Inje Institute of Pharmaceutical Sciences and Research, Inje University, Gimhae 50834, Republic of Korea; pusoon@inje.ac.kr; 4Department of Pharmacy, College of Pharmacy and Research Institute for Drug Development, Pusan National University, Busan 46241, Republic of Korea; hyjung@pusan.ac.kr

**Keywords:** tyrosinase, zebrafish larva, melanin, depigmentation, kojic acid, B16F10 cells

## Abstract

Fifteen compounds (**1**–**15**) constructed on a hybrid structure combining a β-phenyl-α,β-unsaturated carbonyl template and a 2-aminothiazol-4(5*H*)-one scaffold were designed and synthesized as potential novel anti-tyrosinase substances. Two compounds (**10** and **15**) showed more potent inhibition against mushroom tyrosinase than kojic acid, and the inhibitory activity of **10** (IC_50_ value: 1.60 μM) was 11 times stronger than that of kojic acid. Lineweaver–Burk plots indicated that these two compounds were competitive inhibitors that bound to the mushroom tyrosinase active site, which was supported by in silico experiments. Compound **10** was an anti-tyrosinase and anti-melanogenic substance in B16F10 cells and was more potent than kojic acid, without cytotoxicity. Compound **15** exhibited the most potent effect on zebrafish larval depigmentation and showed a depigmentation effect comparable to kojic acid, even at a concentration 200 times lower. Compounds **8** and **10** exhibited strong antioxidant capacities, scavenging 2,2-diphenyl-1-picrylhydrazyl, (2,2-azino-bis-3-ethylbenzothiazoline-6-sulphonic acid)^+^ radicals, and reactive oxygen species. Hybrid compounds **10** and **15** are potential therapeutic agents for skin hyperpigmentation disorders.

## 1. Introduction

Melanin is a polymeric pigment that provides organisms with color and is abundant in various living organisms ranging from bacteria to humans [1]. In humans, excluding neuromelanin, melanin is divided into two types: eumelanin, which has a 5,6-dihydroxyindole structure (black to brown), and pheomelanin, which has a benzothiazine or benzothiazole structure (red to yellow) [2,3,4]. These two melanins greatly influence the determination of skin color. Melanin is biosynthesized in melanocyte organelles called melanosomes and is transferred to keratinocytes, which are abundant in the epidermis [5]. Skin color is mainly determined by the ratio of the two types of melanin within keratinocytes. Melanin plays a role in protecting the skin by absorbing harmful ultraviolet rays [3]. However, excessive melanin production in a specific area of the skin causes hyperpigmentation diseases, such as freckles and melasma, which have a negative view in terms of aesthetics [6,7,8].

Tyrosinase is an oxidizing enzyme that contains two copper ions in its active site and oxidizes substrates l-tyrosine and l-DOPA to l-DOPA and dopaquinone, respectively [9]. Tyrosinase is found in most living organisms, including bacteria, fungi, plants, and animals, and is responsible for the initiation of melanin biosynthesis [10]. As tyrosinase is responsible for the rate-limiting step of melanogenesis, a series of melanin production processes [11,12], it is considered a key target for anti-melanogenic agents. Arbutin, 4-butylresorcinol, kojic acid, tropolone, and hydroquinone are tyrosinase inhibitors, some of which are used clinically as skin-brightening agents [13,14,15,16,17,18]. However, these ingredients used clinically have drawbacks such as weak skin-brightening potency, formulation instability, poor bioavailability, and adverse effects such as contact dermatitis, skin irritation, carcinogenicity, mutagenicity, skin cancer, thyroid cancer, nephrotoxicity, and genotoxicity [18,19,20]. Thus, the demand for new skin-brightening cosmetics and medications for skin hyperpigmentation that are safe and effective for clinical applications still exists.

We demonstrated that a β-phenyl-α,β-unsaturated carbonyl (PUSC) template (Figure 1) plays an essential role in enabling compounds to inhibit tyrosinase activity [21,22,23,24]. Derivatives bearing 2-aminothiazol-4(5*H*)-one (Figure 1) have various biological activities such as anticancer [25,26], antibacterial [26,27,28], antifungal [26,28], antitubercular [29], anti-hepatitis viral [30], selective 11β-hydroxysteroid dehydrogenase type 1 inhibiting [31,32], and type 2 diabetes treatment [33] effects. The hybrid of the 2-aminothiazol-4(5*H*)-one scaffold, which shows various biological activities, and the PUSC template produced are promising anti-melanogenic compounds with strong tyrosinase inhibitory activity (Figure 1). After the hybridization of two components, further modification was performed; a naphthalene ring was introduced to increase the lipophilicity of the target compounds for dermal applications, and various R substituents were introduced to strengthen tyrosinase inhibitory activity (Figure 1). In this study, target compounds **1**–**15** designed in this way were evaluated for their inhibitory activity on the tyrosinase enzyme itself and on mammalian cellular tyrosinase, as well as for their ability to inhibit melanin biosynthesis in mammalian cells. In addition, their ability to inhibit melanogenesis was demonstrated in vivo.

## 2. Results and Discussion

### 2.1. Synthesis of Target Compounds ***1**–**15***

As depicted in Figure 1, 2-(naphthalen-1-ylamino)thiazol-4(5*H*)-one (**17**) was used as the key intermediate for the synthesis of target compounds **1**–**15**. Key intermediate **17** was synthesized via a two-step reaction. 1-Naphthylamine was condensed with chloroacetyl chloride to provide the corresponding amide **16**, which was refluxed in ethyl alcohol in the presence of ammonium thiocyanate to generate **17** [34]. The simple substitution product **16a** was obtained by refluxing α-chloroacetamide **16** with ammonium thiocyanate, which then underwent a ring formation–rearrangement process at reflux temperature, furnishing the rearranged compound **17**. The condensation of key intermediate **17** with various appropriate benzaldehydes in the presence of 3.0 equivalent NaOAc in acetic acid generated the target compounds **1**–**15** as solids having 48–82% yield. The substituents on the final compounds **1**–**15** included hydroxy, methoxy, ethoxy, fluorine, bromine, and *tert*-butyl groups, which either donate (hydroxyl and alkoxy) or withdraw (fluorine and bromine) electrons to the phenyl ring or provide bulkiness (*tert*-butyl). The structures of the target compounds were confirmed by ^1^H and ^13^C nuclear magnetic resonance (NMR) and mass spectroscopy. The alkenes of the target compounds had (*Z*)-geometries. The (*Z*)-geometry is a thermodynamical more stable form than the (*E*)-geometry, owing to the intramolecular hydrogen bond between the H_β_ and carbonyl oxygen and the steric hindrance between the β-phenyl ring and carbonyl [35,36]. In addition, the H_β_ peaks of the PUSC template in NMR spectra for compounds **1**–**15** appeared downfield (7.45–7.83 ppm). This is possible only in the (*Z*)-isomers where the carbonyl is present on the same side as H_β_. The H_β_ peaks in the (*Z*)-isomers shifted downfield owing to the anisotropic effect of the carbonyl group.

### 2.2. Inhibitory Activity of Target Compounds ***1**–**15*** Against Mushroom Tyrosinase

The potency of the synthesized target compounds **1**–**15** in the inhibition of tyrosinase activity was evaluated using a mushroom tyrosinase assay. Inhibition was measured in the presence of l-tyrosine and l-DOPA as substrates. To obtain the half-maximal inhibitory concentration (IC_50_) value, the tyrosinase inhibitory activity was measured at 3–5 different concentrations of each compound.

As depicted in Table 1, compounds **1**, **4**, **5**, and **6**, with 4-methoxy, 2,4-dimethoxy, 4-hydroxy-3-methoxy, and 3-hydroxy-4-methoxy substituents on the β-phenyl ring, respectively, showed strong-to-moderate tyrosinase inhibitory activity in the presence of l-tyrosine (IC_50_ values: 36.95, 43.73, 78.88, and 69.38 μM, respectively) and moderate-to-weak tyrosinase inhibitory activity in the presence of l-DOPA (IC_50_ values: 58.43, 99.99, 44.75, and 136.09 μM, respectively). Kojic acid used as a positive control had strong tyrosinase inhibitory activity (IC_50_ values: 27.41 and 24.09 μM in the presence of l-tyrosine and l-DOPA, respectively). Compounds **12**–**14** substituted in positions 3, 4, and 5 and compounds **2**, **7**, and **8**, with 4-hydroxy, 3-ethoxy-4-hydroxy, and 3,4-dihydroxy on the β-phenyl ring, exhibited weak or no tyrosinase inhibitory activity (IC_50_ values > 145 μM) regardless of the type of substrate used. Compound **3**, bearing a 3,4-dimethoxy substituent, showed different tyrosinase inhibitory activities depending on the substrate type: weak inhibitory activity in the presence of l-DOPA (IC_50_ value: 163.60 μM) and strong inhibitory activity in the presence of l-tyrosine (IC_50_ value: 28.72 μM), similar to kojic acid. Compound **8**, bearing a 3,4-dihydroxy substituent, had no tyrosinase inhibitory activity (IC_50_ value > 200 μM); however, exchange with a 3,4-difluoro substituent increased the tyrosinase inhibitory activity (IC_50_ values of **9**: 60.02 and 57.86 μM in the presence of l-DOPA and l-tyrosine, respectively). Compound **11**, bearing a 2,4-difluoro substituent, exhibited moderate tyrosinase inhibition (IC_50_ values: 57.76 and 62.61 μM in the presence of l-DOPA and l-tyrosine, respectively) similar to **9** with a 3,4-difluoro substituent. However, exchange with a 2,4-dihydroxy substituent greatly enhanced its tyrosinase inhibitory activity (IC_50_ values of **10**: 1.60 and 2.86 μM in the presence of l-tyrosine and l-DOPA, respectively). The tyrosinase inhibition was increased 39 and 20 times in the presence of l-tyrosine and l-DOPA, respectively, and these IC_50_ values were considerably lower than those of kojic acid. Compound **15** with a 3-bromo-4-hydroxy substituent also showed stronger inhibition with IC_50_ values of 18.09 and 6.92 μM in the presence of l-tyrosine and l-DOPA, respectively, than kojic acid.

The structure–activity relationships (SARs) of compounds **1**–**15** are shown in Figure 2. Compound **2**, with a 4-hydroxy group, showed weak-to-nil mushroom tyrosinase inhibitory activity, and the addition of an additional substituent at position 3 increased the inhibition in the order of Br, OMe, and OH; compound **15**, with a 3-bromo-4-hydroxy group, had a stronger inhibitory potency than kojic acid (Figure 2A). The addition of the same two groups, methoxy and *t*-butyl, at positions 3 and 5 of compound **2** did not show inhibition activity (IC_50_ values > 150 μM). The insertion of an additional OH group at position 2 of compound **2** dramatically enhanced the tyrosinase inhibitory activity and lowered the IC_50_ value close to the nanomolar concentration. Contrastingly, compound **1**, with a 4-methoxy group, exhibited strong-to-moderate tyrosinase inhibitory activity, and the introduction of an additional substituent at position 3 increased its tyrosinase inhibition in the order OMe, OH, and OEt in the presence of l-tyrosine (Figure 2B). However, the order changed in the presence of l-DOPA: OH induced the strongest tyrosinase inhibitory activity, followed by OMe and OEt. The addition of two identical groups (OMe) at positions 3 and 5 of compound **1** decreased tyrosinase inhibition, with an IC_50_ value of > 200 μM. However, the insertion of OMe at position 2 maintained the tyrosinase inhibitory activity. The insertion of two fluorine atoms at positions 2 and 4 or 3 and 4 on the β-phenyl ring exhibited strong-to-moderate tyrosinase inhibition with approximate IC_50_ values of 60 μM regardless of the type of substrate used (**9** and **11**; Figure 2C). These SAR results suggest that substitutions at positions 2 and 4 of the β-phenyl ring significantly affect tyrosinase inhibition and that these substituents are preferably hydrogen bond donors such as OMe or F. However, OH, which can function as both a hydrogen bond donor and acceptor, is considerably better.

We compared these SAR results with those obtained for (*E*)-benzylidene-1-indanone (BID) derivatives [37]. Just as the addition of a 2-hydroxy substituent to the BID compound with a 4-hydroxy substituent strongly increased its tyrosinase inhibitory activity (IC_50_ value: 39.7 μM vs. 0.03 μM), compound **10**, which was obtained by adding a 2-hydroxy substituent to compound **2** with a 4-hydroxy substituent, strongly increased the tyrosinase inhibitory activity (IC_50_ value: 148.1 μM vs. 2.9 μM). When bromine was added to the BID compound containing a 4-hydroxy substituent, the tyrosinase inhibitory activity decreased (IC_50_ value: 39.7 μM vs. >200 μM), but when bromine was added to compound **2** with a 4-hydroxy substituent, the inhibitory activity increased (IC_50_ value: 148.1 μM vs. 6.9 μM). As in many studies [22,24,38], the BID compound with a 4-hydroxy substituent showed stronger tyrosinase inhibitory activity than the BID compound with a 4-methoxy substituent (IC_50_ value: 39.7 μM vs. >200 μM), but compound **2** with a 4-hydroxy substituent showed weaker tyrosinase inhibitory activity than compound **1** with a 4-methoxy substituent (IC_50_ value: 148.1 μM vs. 58.4 μM).

### 2.3. Kinetic Analysis of Mushroom Tyrosinase in the Presence of Compound ***10*** or ***15***

The inhibition mechanism of compounds **10** and **15**, which demonstrated more potent tyrosinase inhibition than kojic acid, was determined by kinetic studies of mushroom tyrosinase using Lineweaver–Burk plots. The plots for inhibitors **10** and **15** were obtained by measuring the initial rate of dopachrome production in the presence of various l-DOPA substrate concentrations (Figure 3A,C). For the kinetic experiments, compounds **10** and **15** were used at 0, 1, 2, and 4 μM and at 0, 4, 8, and 16 μM, respectively. Each Lineweaver–Burk plot showed that the maximal velocity of the dopachrome production rate was constant regardless of the inhibitor concentration. The four lines in each plot meet at one point on the *y*-axis, indicating that compounds **10** and **15** are inhibitors that compete with the substrates for binding to the active site of tyrosinase. The Michaelis constant (K_m_) values of these competitive inhibitors increased with increasing inhibitor concentration.

K_i_, known as the dissociation constant or inhibition constant, describes the binding affinity between the enzyme and inhibitor. To examine the binding affinity between tyrosinase and compounds **10** and **15**, the K_i_ values of each compound were obtained by converting Lineweaver–Burk plots into the corresponding Dixon plots. Dixon plots were acquired by plotting the inverse of the initial dopachrome formation rate against the inhibitor concentration (Figure 3B,D). Each Dixon plot produces four lines that merge at one point in the second quadrant. The absolute value of the x co-ordinate of the merge point represents the K_i_ value. The K_i_ values of **10** and **15** were 2.04 × 10^−6^ and 4.87 × 10^−6^ M, respectively. The lower the K_i_ value, the stronger the binding affinity. Thus, compound **10** bound more strongly to the tyrosinase active site than compound **15**.

### 2.4. Docking Simulation

Compounds **10** and **15** strongly and competitively inhibited mushroom tyrosinase. The binding affinities and pharmacophores of these compounds were investigated using the three-dimensional (3D) X-ray structure of *Agaricus bisporus* mushroom tyrosinase (PDB ID: 2Y9X), AutoDock Vina, and LigandScout. The test ligands (kojic acid [positive control], **10**, and **15**) were docked with the 3D tyrosinase protein.

Figure 4 shows the docking results of these ligands in 3D and 2D modes. Compound **10** interacted with five amino acid residues present in the tyrosinase active site. Val248 hydrophobically interacted with the β-phenyl ring, whereas three amino acids (Ala286, Phe264, and Val283) hydrophobically interacted with the naphthalene ring. The 2-hydroxyl group of the β-phenyl ring formed a hydrogen bond with His244 and served as a hydrogen bond donor. These chemical interactions between compound **10** and tyrosinase provided **10** with a strong binding energy of −8.4 kcal/mol. Compound **15** also bound tightly to the active site of tyrosinase through several interactions. The bromine and naphthalene rings participated in hydrophobic interactions; bromine interacted with Asn260, and the naphthalene ring interacted with three amino acids, Val283, Ala286, and Phe264. Compound **15** also participated in hydrogen bonding. The 2-amino group acted as a hydrogen bond donor and formed a hydrogen bond with Asn260. The 4-hydroxyl group of the β-phenyl ring also created two hydrogen bonds with Glu322 and His85, acting as a hydrogen bond donor in both hydrogen bonds. These chemical interactions resulted in **15** having a strong binding energy of −8.9 kcal/mol. Meanwhile, kojic acid created two interactions: a hydrogen bond between Asn260 and the 2-hydroxymethyl and pi-pi stacking between His263 and 4-pyranone, providing a binding energy of −5.4 kcal/mol. Thus, compounds **10** and **15** bind to the tyrosinase active site more strongly than kojic acid.

Compounds **10** and **15** hydrophobically interacted with the same three amino acid residues (Phe264, Val283, and Ala286) in their naphthalene rings. In addition, compounds **10** and **15** increased binding affinity by allowing the hydroxyl group at position 2 and position 4 of the phenyl ring to participate in hydrogen bonding, respectively. Kojic acid is known to inhibit the activity of tyrosinase by chelating the copper of tyrosinase using the α-hydroxy ketone. Compounds **10** and **15** do not have a chemical structure that can chelate copper. These compounds were designed to bind well to the surrounding amino acids at the active site of tyrosinase. Therefore, as shown in the docking simulation results, kojic acid and these compounds had very different interaction patterns with tyrosinase amino acid residues.

### 2.5. Cytotoxicity on B16F10 Cells

Compounds **10** and **15** potently inhibited mushroom tyrosinase activity. Therefore, we investigated whether these compounds exert tyrosinase inhibitory activity in B16F10 mammalian cells. We examined the effects of these compounds on B16F10 cell viability. Cell viability was assessed for 72 h at five concentrations (1, 2, 5, 10, and 20 μM) of compounds **10** and **15**.

Compound **10** had no perceptible effect on B16F10 cell viability up to a maximum concentration of 20 μM, whereas compound **15** clearly reduced cell viability in a concentration-dependent manner (Figure 5). Even at 1 μM, significant cytotoxicity was observed. Thus, the use of compound **15** in further experiments using B16F10 cells was excluded.

### 2.6. Effects on Melanogenesis and Cellular Tyrosinase Activity in B16F10 Cells

Compound **10** potently exerted mushroom tyrosinase inhibitory activity and showed no cytotoxicity to B16F10 cells up to 20 μM. Thus, we investigated whether **10** could exert this potency in inhibiting melanin biosynthesis in B16F10 mammalian cells. B16F10 cells were pre-treated with **10** at three concentrations (5, 10, and 20 μM) or with kojic acid (20 μM) as a positive control for 1 h and then treated with stimulators (1 μM α-melanocyte-stimulating hormone [α-MSH] and 200 μM 3-isobutyl-1-methylxanthine [IBMX]). After 72 h, the effect of compound **10** on melanin biosynthesis was determined.

Exposure to these stimulators significantly increased melanin biosynthesis by 3.9-fold (Figure 6A). Treatment with kojic acid weakly reduced the stimulator-induced melanin content by 3.4-fold. Treatment with **10** strongly reduced stimulator-induced melanin content dose-dependently. Even at 5 μM, **10** reduced the stimulator-induced melanin content levels similarly to kojic acid. At 20 μM, compound **10** (86%) greatly reduced melanin content levels below those of controls (100%).

We analyzed the effect of **10** on B16F10 cellular tyrosinase activity to determine whether the ability of **10** inhibiting melanogenesis results from its ability to inhibit cellular tyrosinase activity. Similar to the melanin content experiments, B16F10 cells were pre-exposed to compound **10** (5, 10, and 20 μM) or kojic acid (20 μM) for 1 h and then exposed to the same concentrations of stimulants (α-MSH and IBMX) as used for the melanin content experiments. After 72 h, the influence of **10** and kojic acid on cellular tyrosinase inhibitory activity was determined.

As shown in Figure 6B, exposure to stimulators increased cellular tyrosinase activity by 3.9-fold compared to the untreated control, and kojic acid treatment weakly reduced the increased cellular tyrosinase activity by 3.3-fold. Treatment with compound **10** induced a dose-dependent decrease in cellular tyrosinase activity. Compound **10** at 10 μM showed slightly more potent tyrosinase inhibitory potency than kojic acid at 20 μM, and at 20 μM, it reduced cellular tyrosinase activity to almost the level of cellular tyrosinase activity in the untreated control. Compared with the cellular tyrosinase activity levels at the same concentration (20 μM), compound **10** exerted a much stronger cellular tyrosinase inhibitory effect than kojic acid. A comparison of the melanin content and cellular tyrosinase activity indicated that the anti-melanogenic efficacy of **10** was closely related to its ability to inhibit cellular tyrosinase activity.

### 2.7. Effect on In Situ Cellular Tyrosinase Activity in B16F10 Cells

The effect of **10** on cellular tyrosinase activity was investigated in situ using l-DOPA and B16F10 cells. This experimental method allows for the direct observation of cellular melanin production. B16F10 cells were pre-treated with **10** (5, 10, and 20 μM) or kojic acid (20 μM; a positive control) for 1 h, and then exposed to stimulators (1 μM α-MSH and 200 μM IBMX). After 72 h of incubation, the cells were treated with l-DOPA for 2 h to induce melanin production.

As depicted in Figure 7, the group treated with only stimulators showed increased melanin production compared with the untreated control. When kojic acid was used, melanin production was observed in some cells, although melanin production was suppressed compared with that in the group treated with only stimulators. Compound **10** treatment dose-dependently decreased melanin formation, and at 5 μM, it exhibited a similar potency to kojic acid at 20 μM. Compound **10** at 20 μM reduced melanin formation to the level of the untreated control. These results suggest that compound **10** is a promising candidate for the treatment of skin hyperpigmentation disorders.

### 2.8. In Vivo Depigmentation Effect in Zebrafish Larvae

Zebrafish embryos were used to determine whether the synthesized compounds could inhibit melanin synthesis in vivo. First, through preliminary experiments in which the compounds were treated at a concentration of 0.1 mM, compound **15** was selected for the main in vivo experiments. Zebrafish embryos obtained through natural mating were dechorionated 24 h post-fertilization (24 hpf) and treated with compound **15** at concentrations of 0.1 and 0.3 mM at 28 hpf (Figure 8A). The degree of depigmentation in zebrafish larvae was confirmed by photography 48 h later (Figure 8A). The density of the zebrafish larvae was measured using a CS analyzer 3.2 (Atto, Tokyo, Japan). Kojic acid (20 mM) was used to compare depigmentation activity.

Photographs confirmed that the pigmentation of zebrafish larvae was reduced in the kojic acid-treated group compared with that in the control group (Figure 8B). Compound **15** showed a depigmentation capacity similar to that of kojic acid, even at a concentration 200 times lower. Pigmentation density in the dorsal and lateral views showed similar results (Figure 8C). On the other hand, the tyrosinase activity of zebrafish larvae was investigated at 76 hpf. Compared with the control group, the kojic acid-treated group showed significantly reduced tyrosinase activity (Figure 8D). Compound **15** also decreased tyrosinase activity in a concentration-dependent manner, inhibiting the tyrosinase activity of zebrafish larvae at a level comparable to that of kojic acid, even at a concentration 70 times lower than that of kojic acid.

### 2.9. Antioxidant Ability of Compounds ***1**–**15***

#### 2.9.1. 2,2-Diphenyl-1-picrylhydrazyl (DPPH) Radical-Scavenging Ability of Compounds **1**–**15**

To evaluate the DPPH radical-scavenging efficacy of **1**–**15**, a DPPH methanol solution was mixed with compounds **1**–**15**, and the mixture was left in the dark. After 30 min, the radical-scavenging efficacy of each compound was determined by measuring the optical density at 517 nm. Vitamin C (l-ascorbic acid) was used for comparing radical-scavenging activity, and all test samples were treated at 500 μM.

The results of the DPPH radical-scavenging activity assay are presented in Figure 9A. Vitamin C scavenged 90% of the DPPH radicals, and compounds **1**–**15** showed different DPPH radical-scavenging efficacies depending on their chemical structures. Compounds bearing a 3-alkoxy-4-hydroxy group (**5** and **7**) showed strong scavenging efficacies (62% and 61% scavenging, respectively). Exchange of the methoxy and hydroxyl groups of **5** reduced the DPPH radical-scavenging activity (**6**: 21% scavenging). Compounds with 3,5-dialkoxy-4-hydroxyl or 3,5-di-*tert*-butyl-4-hydroxyl groups (**13** and **14**) exhibited strong DPPH radical-scavenging abilities, with 51% and 68% scavenging activities, respectively. Out of the synthesized compounds, compound **8**, which has a 3,4-dihydroxyl group, showed the strongest DPPH radical-scavenging efficacy, with 77% scavenging activity. Compound **10**, which has a 2,4-dihydroxyl group and showed the most potent mushroom tyrosinase inhibitory activity and anti-melanogenic activity in B16F10 cells, exhibited moderate DPPH radical-scavenging ability (51%).

#### 2.9.2. 2,2′-Azino-bis(3-ethylbenzothiazoline-6-sulfonic acid) (ABTS) Radical Cation-Scavenging Ability of Compounds **1**–**15**

The ABTS^+^-scavenging efficacy of compounds **1**–**15** was determined by oxidizing ABTS with K_2_S_2_O_8_ to prepare ABTS^+^. The prepared ABTS^+^ solution was mixed with the test samples (**1**–**15** and Trolox, a positive control) and the radical-scavenging efficacy of the test samples was explored by measuring the decrease in optical density at 732 nm. All test samples were tested at 100 μM.

Among compounds **1**–**15**, eight compounds showed strong ABTS^+^-scavenging efficacy with > 80% ABTS^+^ inhibition (Figure 9B). Trolox scavenged ABTS^+^ by 99%, and four compounds, **8** with 3,4-dihydroxyphenyl, **10** with 2,4-dihydroxyphenyl, **13** with 4-hydroxy-3,5-dimethoxyphenyl, and **14** with 3,5-di-*tert*-butyl-4-hydroxyphenyl, strongly scavenged ABTS^+^ by 99, 97, 97, and 95%, respectively, at levels which were comparable to those of Trolox.

The antioxidant activity of compounds contributes to the inhibition of melanogenesis [39,40]. Thus, the ability of compound **10** to inhibit melanin biosynthesis in B16F10 cells may be due in part to its antioxidant capacity, as well as its direct inhibition of cellular tyrosinase.

#### 2.9.3. Reactive Oxygen Species (ROS)-Scavenging Ability of **1**–**15**

To assess the ability of compounds **1**–**15** to scavenge ROS, 2′,7′-dichlorodihydrofluorescein (DCFH) solution, which was obtained from hydrolysis of DCFH diacetate by esterase, was mixed with 3-morpholinosydnonimine (SIN-1), a ROS generator, and test samples (**1**–**15** and Trolox, a positive material). ROS produces 2′,7′-dichlorofluorescein (DCF), a fluorescent material, by oxidizing DCFH. The ROS-scavenging activity of the samples was determined by measuring the fluorescence of DCF. The results were obtained at 10 μM SIN-1 and 40 μM sample.

Trolox showed strong ROS-scavenging activity, and six of the compounds among **1**–**15** significantly inhibited ROS levels (Figure 9C). In particular, compound **8**, bearing a 3,4-dihydroxyphenyl ring, exhibited a very strong ROS-scavenging capacity, which was more potent than that of Trolox.

To summarize the structures important for antioxidant activity, in general, the greater the number of hydroxyl groups on the phenyl ring, the stronger the antioxidant activity that was exhibited. In particular, the catechol (3,4-dihydroxyphenyl) structure showed strong antioxidant activity in various antioxidant experiments.

## 3. Materials and Methods

### 3.1. Synthesis

#### 3.1.1. General Methods of Synthesis

Solvents were purchased from DaeJung (Siheung-si, Republic of Korea), and all chemicals were obtained from commercial sources (SEJIN CI Co., Ltd. (Seoul, Republic of Korea) and Thermo Fisher Scientific (Waltham, MA, USA)). Low-resolution mass (LRMS) data were acquired in electrospray ionization (ESI) positive or negative mode on an Expression CMS spectrometer (Advion Interchim Scientific, Ithaca, NY, USA), and high-resolution mass (HRMS) data were acquired on ZenoTOF 7600 mass spectrometer (SCIEX, Framingham, MA, USA) ^13^C and ^1^H NMR spectra were recorded on a JEOL ECZ400S instrument (JEOL Ltd., Tokyo, Japan) or a Varian Unity AS500 spectrometer (Agilent Technologies, Santa Clara, CA, USA). The *J* values and chemical shifts obtained from the NMR data are presented in Hz and ppm, respectively. NMR peak-splitting patterns: s, singlet; br, broad singlet; t, triplet; d, doublet; dd, doublet of doublets; m, multiplet.

#### 3.1.2. Synthesis of 2-Chloro-*N*-(naphthalen-1-yl)acetamide (**16**)

Chloroacetyl chloride (5.6 mL, 70.41 mmol) was added dropwise to a solution of 1-naphthylamine (5.0 g, 34.92 mmol) in acetone (100 mL), and the reaction mixture was stirred at room temperature for 6 h. After evaporating the acetone and adding water (20 mL), a saturated NaHCO_3_ solution was added to adjust the pH to 7. The precipitate was filtered and successively washed with water and hexane to obtain **16** (5.858 g, 76%).

^1^H NMR (CDCl_3_, 500 MHz) δ 8.77 (brs, 1H, NH), 7.99 (d, 1H, *J* = 8.0 Hz), 7.90 (d, 1H, *J* = 8.0 Hz), 7.87 (d, 1H, *J* = 8.0 Hz), 7.75 (d, 1H, *J* = 8.0 Hz), 7.58 (t, 1H, *J* = 8.0 Hz), 7.54 (t, 1H, *J* = 8.0 Hz), 7.51 (t, 1H, *J* = 8.0 Hz), 4.35 (s, 2H, CH_2_).

#### 3.1.3. Synthesis of 2-(Naphthalen-1-ylamino)thiazol-4(5*H*)-one (**17**)

A solution of **16** (5.415 g, 24.65 mmol) in ethanol (EtOH, 50 mL) was refluxed in the presence of NH_4_SCN (3.75 g, 49.26 mmol) for 24 h. After evaporation, the residue was filtered and washed several times with water and hexane to obtain **17** (4.857 g, 81%).

^1^H NMR (dimethyl sulfoxide [DMSO]-*d*_6_, 400 MHz) δ 12.00 (brs, 1H, NH), 7.93–7.88 (m, 2H), 7.70 (d, 1H, *J* = 8.4 Hz), 7.56–7.44 (m, 3H, 3-H, 6-H, 7-H), 7.04 (d, 1H, *J* = 7.2 Hz), 3.95 (s, 2H, CH_2_); ^13^C NMR (DMSO-*d*_6_, 125 MHz) δ 175.2, 158.7, 144.5, 134.4, 128.4, 127.6, 126.9, 126.4, 126.2, 124.8, 123.4, 116.4, 34.8; LRMS (ESI+) *m*/*z* 243 (M + H)^+^, 265 (M + Na)^+^.

#### 3.1.4. General Procedure for the Synthesis of Compounds **1**–**15**

A solution of **17** (100 mg, 0.41 mmol) and the appropriate benzaldehyde (1.0 equiv.) in acetic acid (2–3 mL) was refluxed in the presence of NaOAc (3.0 equiv.) for 2–8 d. The target compounds were purified using the following three methods: (1) washing several times with water; (2) washing with water and diethyl ether; or (3) washing with water, followed by column chromatography of the filter cake using dichloromethane/methanol (20–50:1) as the eluent. The title compounds **1**–**15** were obtained as solids in yields of 48–82%.

*(Z)-5-(4-Methoxybenzylidene)-2-(naphthalen-1-ylamino)thiazol-4(5H)-one* (**1**) [41]. ^1^H NMR (DMSO-*d*_6_, 500 MHz) δ 12.50 (s, 1H, NH), 7.94–7.90 (m, 2H, 5-H, 8-H), 7.73 (d, 1H, *J* = 8.0 Hz, 4-H), 7.59 (s, 1H, vinylic H), 7.55–7.47 (m, 3H, 3-H, 6-H, 7-H), 7.37 (d, 2H, *J* = 9.0 Hz, 2′-H, 6′-H), 7.12 (d, 1H, *J* = 7.0 Hz, 2-H), 6.95 (d, 2H, *J* = 9.0 Hz, 3′-H, 5′-H), 3.72 (s, 3H, OCH_3_); ^13^C NMR (DMSO-*d*_6_, 125 MHz) δ 168.6, 160.9, 152.4, 144.9, 134.4, 132.1, 130.0, 128.4, 127.5, 127.1, 126.6, 126.5, 126.3, 125.2, 123.5, 120.6, 116.5, 115.2, 55.8; LRMS (ESI+) *m*/*z* 383 (M + Na)^+^; LRMS (ESI−) *m*/*z* 359 (M − H)^−^; yield: 76%.

*(Z)-5-(4-Hydroxybenzylidene)-2-(naphthalen-1-ylamino)thiazol-4(5H)-one* (**2**) [41]. ^1^H NMR (DMSO-*d*_6_, 500 MHz) δ 12.22 (brs, 1H, NH), 10.11 (s, 1H, OH), 7.94 (d, 1H, *J* = 8.5 Hz), 7.91 (d, 1H, *J* = 8.5 Hz), 7.74 (d, 1H, *J* = 8.5 Hz), 7.56–7.48 (m, 3H, 3-H, 6-H, 7-H), 7.54 (s, 1H, vinylic H), 7.28 (d, 2H, *J* = 9.0 Hz, 2′-H, 6′-H), 7.12 (d, 1H, *J* = 7.0 Hz, 2-H), 6.79 (d, 2H, J = 9.0 Hz, 3′-H, 5′-H); ^13^C NMR (DMSO-*d*_6_, 125 MHz) δ 168.5, 159.8, 152.6, 144.8, 134.4, 132.4, 130.5, 128.4, 127.5, 127.0, 126.6, 126.5, 125.1, 124.6, 123.4, 119.1, 116.6, 116.5; LRMS (ESI−) *m*/*z* 345 (M − H)^−^; yield: 79%.

*(Z)-5-(3,4-Dimethoxybenzylidene)-2-(naphthalen-1-ylamino)thiazol-4(5H)-one* (**3**). ^1^H NMR (DMSO-*d*_6_, 500 MHz) δ 12.50 (s, 1H, NH), 7.94–7.91 (m, 2H, 5-H, 8-H), 7.73 (d, 1H, *J* = 8.0 Hz, 4-H), 7.60 (s, 1H, vinylic H), 7.55–7.47 (m, 3H, 3-H, 6-H, 7-H), 7.13 (d, 1H, *J* = 7.0 Hz, 2-H), 7.11 (s, 1H, 2′-H), 6.94 (d, 1H, *J* = 8.0 Hz, 5′-H), 6.91 (d, 1H, *J* = 8.0 Hz, 6′-H), 3.70 (s, 3H, OCH_3_), 3.68 (s, 3H, OCH_3_); ^13^C NMR (DMSO-*d*_6_, 125 MHz) δ 168.4, 152.0, 150.7, 149.2, 144.8, 134.4, 130.4, 128.4, 127.6, 127.1, 126.5, 126.5, 125.2, 123.5, 122.6, 120.8, 116.5, 114.7, 112.5, 56.0, 56.0; LRMS (ESI−) *m*/*z* 389 (M − H)^−^; yield: 71%.

*(Z)-5-(2,4-Dimethoxybenzylidene)-2-(naphthalen-1-ylamino)thiazol-4(5H)-one* (**4**). ^1^H NMR (DMSO-*d*_6_, 500 MHz) δ 7.92–7.90 (m, 2H, 5-H, 8-H), 7.81 (s, 1H, vinylic H), 7.71 (d, 1H, *J* = 7.5 Hz, 4-H), 7.54–7.45 (m, 3H, 3-H, 6-H, 7-H), 7.10 (d, 1H, *J* = 8.0 Hz, 2-H), 7.08 (d, 1H, *J* = 9.0 Hz, 6′-H), 6.58 (d, 1H, *J* = 1.5 Hz, 3′-H), 6.49 (dd, 1H, *J* = 9.0, 1.5 Hz, 5′-H), 3.82 (s, 3H, OCH_3_), 3.72 (s, 3H, OCH_3_); ^13^C NMR (DMSO-*d*_6_, 125 MHz) δ 168.5, 162.9, 159.9, 152.4, 144.9, 134.4, 129.8, 128.4, 127.6, 127.0, 126.5, 126.5, 125.1, 124.4, 123.5, 120.1, 116.5, 115.1, 106.6, 98.9, 56.3, 55.9; HRMS (EDA) *m*/*z* C_22_H_19_N_2_O_3_S (M + H)^+^ calcd 391.1111, obsd 391.1111; yield: 82%.

*(Z)-5-(4-Hydroxy-3-methoxybenzylidene)-2-(naphthalen-1-ylamino)thiazol-4(5H)-one* (**5**) [41]. ^1^H NMR (DMSO-*d*_6_, 500 MHz) δ 12.21 (brs, 1H, NH), 9.74 (s, 1H, OH), 7.94–7.91 (m, 2H, 5-H, 8-H), 7.74 (d, 1H, *J* = 8.5 Hz, 4-H), 7.57 (s, 1H, vinylic H), 7.56–7.48 (m, 3H, 3-H, 6-H, 7-H), 7.13 (d, 1H, *J* = 7.5 Hz, 2-H), 7.10 (s, 1H, 2′-H), 6.83 (d, 1H, *J* = 8.0 Hz, 6′-H), 6.80 (d, 1H, *J* = 8.0 Hz, 5′-H), 3.71 (s, 3H, OCH_3_); ^13^C NMR (DMSO-*d*_6_, 125 MHz) δ 168.4, 152.0, 149.3, 148.2, 144.8, 134.3, 130.8, 128.4, 127.6, 127.0, 126.5, 126.4, 125.2, 123.5, 123.0, 119.6, 116.6, 116.4, 115.5, 56.1; LRMS (ESI−) *m*/*z* 375 (M − H)^−^; yield: 80%.

*(Z)-5-(3-Hydroxy-4-methoxybenzylidene)-2-(naphthalen-1-ylamino)thiazol-4(5H)-one* (**6**). ^1^H NMR (CDCl_3_, 500 MHz) δ 7.92 (d, 1H, *J* = 8.5 Hz), 7.85 (d, 1H, *J* = 7.5 Hz), 7.71 (d, 1H, *J* = 8.5 Hz), 7.65 (s, 1H, vinylic H), 7.50 (t, 1H, *J* = 7.5 Hz), 7.47–7.42 (m, 2H), 7.16 (d, 1H, *J* = 7.5 Hz, 4-H), 6.95 (d, 1H, *J* = 1.5 Hz, 2′-H), 6.91 (dd, 1H, *J* = 8.5, 1.5 Hz, 6′-H), 6.78 (d, 1H, *J* = 8.5 Hz, 5′-H), 5.29 (s, 1H, OH), 3.83 (s, 3H, OCH_3_); ^13^C NMR (DMSO-*d*_6_, 125 MHz) δ 168.4, 153.0, 150.0, 147.3, 145.0, 134.4, 130.5, 128.5, 127.5, 127.1, 126.6, 126.5, 126.4, 125.2, 123.7, 123.5, 120.2, 116.4, 115.6, 112.8, 56.0; LRMS (ESI−) *m*/*z* 375 (M − H)^−^; yield: 81%.

*(Z)-5-(3-Ethoxy-4-hydroxybenzylidene)-2-(naphthalen-1-ylamino)thiazol-4(5H)-one* (**7**). ^1^H NMR (DMSO-*d*_6_, 500 MHz) δ 12.46 (brs, 1H, NH), 9.67 (s, 1H, OH), 7.94–7.91 (m, 2H, 5-H, 8-H), 7.74 (d, 1H, *J* = 8.5 Hz, 4-H), 7.56 (s, 1H, vinylic H), 7.55–7.48 (m, 3H, 3-H, 6-H, 7-H), 7.13 (d, 1H, *J* = 7.0 Hz, 2-H), 7.08 (s, 1H, 2′-H), 6.84–6.79 (m, 2H, 5′-H, 6′-H), 3.96 (q, 2H, *J* = 7.0 Hz, CH_2_CH_3_), 1.26 (t, 3H, *J* = 7.0 Hz, CH_2_CH_3_); ^13^C NMR (DMSO-*d*_6_, 125 MHz) δ 168.5, 151.9, 149.6, 147.4, 144.9, 134.4, 130.9, 128.4, 127.6, 127.0, 126.5, 126.5, 125.2, 123.5, 123.1, 119.4, 116.7, 116.7, 116.4, 64.4, 15.0; LRMS (ESI−) *m*/*z* 389 (M − H)^−^; yield: 68%.

*(Z)-5-(3,4-Dihydroxybenzylidene)-2-(naphthalen-1-ylamino)thiazol-4(5H)-one* (**8**). ^1^H NMR (DMSO-*d*_6_, 500 MHz) δ 12.39 (brs, 1H, NH), 9.62 (s, 1H, OH), 9.34 (s, 1H, OH), 7.93 (d, 1H, *J* = 8.0 Hz), 7.91 (d, 1H, *J* = 8.0 Hz), 7.74 (d, 1H, *J* = 8.5 Hz, 4-H), 7.56–7.48 (m, 3H, 3-H, 6-H, 7-H), 7.45 (s, 1H, vinylic H), 7.12 (d, 1H, *J* = 7.0 Hz, 2-H), 6.83–6.79 (m, 2H, 2′-H, 5′-H), 6.75 (d, 1H, *J* = 8.5 Hz, 6′-H); ^13^C NMR (DMSO-*d*_6_, 125 MHz) δ 168.6, 152.4, 148.4, 146.1, 144.8, 134.3, 130.9, 128.4, 127.4, 127.1, 126.6, 126.5, 125.1, 125.0, 124.1, 123.4, 118.9, 116.6, 116.5, 116.1; LRMS (ESI+) *m*/*z* 363 (M + H)^+^, 385 (M + Na)^+^; yield: 48%.

*(Z)-5-(3,4-Difluorobenzylidene)-2-(naphthalen-1-ylamino)thiazol-4(5H)-one* (**9**). ^1^H NMR (DMSO-*d*_6_, 500 MHz) δ 12.72 (s, 1H, NH), 7.94–7.90 (m, 2H, 5-H, 8-H), 7.74 (d, 1H, *J* = 8.5 Hz, 4-H), 7.61 (s, 1H, vinylic H), 7.58–7.47 (m, 4H, 3-H, 6-H, 7-H, 6′-H), 7.43 (dt, 1H, *J* = 10.5, 8.5 Hz, 5′-H), 7.25–7.21 (m, 1H, 2′-H), 7.12 (d, 1H, *J* = 7.0 Hz, 2-H); ^13^C NMR (DMSO-*d*_6_, 125 MHz) δ 168.1, 150.2 (dd, *J* = 149.9, 12.5 Hz), 149.9 (dd, *J* = 245.4, 12.9 Hz), 144.6, 134.3, 131.6 (dd, *J* = 6.4, 3.8 Hz), 128.4, 127.8, 127.5, 127.1, 126.6 (d, *J* = 2.9 Hz), 126.6, 126.5, 125.4, 125.0, 123.4, 119.5 (d, *J* = 17.6 Hz), 118.8 (d, *J* = 17.8 Hz), 116.5; LRMS (ESI+) *m*/*z* 367 (M + H)^+^, 389 (M + Na)^+^; LRMS (ESI−) *m*/*z* 365 (M − H)^−^; yield: 79%.

*(Z)-5-(2,4-Dihydroxybenzylidene)-2-(naphthalen-1-ylamino)thiazol-4(5H)-one* (**10**). ^1^H NMR (DMSO-*d*_6_, 500 MHz) δ 12.35 (s, 1H, NH), 10.27 (s, 1H, OH), 9.96 (s, 1H, OH), 7.94–7.89 (m, 2H, 5-H, 8-H), 7.83 (s, 1H, vinylic H), 7.72 (d, 1H, *J* = 8.0 Hz, 4-H), 7.55–7.47 (m, 3H, 3-H, 6-H, 7-H), 7.11 (d, 1H, *J* = 7.0 Hz, 2-H), 6.93 (d, 1H, *J* = 9.0 Hz, 6′-H), 6.34 (d, 1H, *J* = 1.5 Hz, 3′-H), 6.24 (dd, 1H, *J* = 9.0, 1.5 Hz, 5′-H); ^13^C NMR (DMSO-*d*_6_, 125 MHz) δ 168.7, 161.4, 159.3, 152.7, 145.0, 134.3, 130.0, 128.4, 127.5, 127.0, 126.6, 126.4, 125.6, 124.9, 123.5, 117.1, 116.4, 112.4, 108.5, 102.9; LRMS (ESI+) *m*/*z* 363 (M + H)^+^, 385 (M + Na)^+^; HRMS (EDA) *m*/*z* C_20_H_15_N_2_O_3_S (M + H)^+^ calcd 363.0798, obsd 363.0802; yield: 63%.

*(Z)-5-(2,4-Difluorobenzylidene)-2-(naphthalen-1-ylamino)thiazol-4(5H)-one* (**11**). ^1^H NMR (DMSO-*d*_6_, 500 MHz) δ 12.75 (brs, 1H, NH), 7.92 (d, 1H, *J* = 8.5 Hz), 7.87 (d, 1H, *J* = 7.5 Hz), 7.69 (d, 1H, *J* = 8.5 Hz, 4-H), 7.55 (s, 1H, vinylic H), 7.51–7.43 (m, 3H, 3-H, 6-H, 7-H), 7.26–7.21 (m, 2H, 3′-H, 6′-H), 7.08 (d, 1H, *J* = 7.0 Hz, 2-H), 6.95 (td, 1H, *J* = 8.5, 2.0 Hz, 5′-H); ^13^C NMR (DMSO-*d*_6_, 125 MHz) δ 167.9, 163.3 (dd, *J* = 250.6, 12.8 Hz), 161.1 (dd, *J* = 252.8, 12.6 Hz), 151.3, 144.6, 134.3, 130.6 (dd, *J* = 10.1, 2.9 Hz), 128.4, 127.5, 127.1, 126.6, 126.5, 126.2, 125.4, 123.4, 119.9 (d, *J* = 5.4 Hz), 118.5 (dd, *J* = 12.0, 3.6 Hz), 116.5, 113.2 (dd, *J* = 21.8, 3.3 Hz), 105.3 (t, *J* = 26.0 Hz); LRMS (ESI+) *m*/*z* 389 (M + Na)^+^; LRMS (ESI−) *m*/*z* 365 (M − H)^−^; yield: 80%.

*(Z)-2-(Naphthalen-1-ylamino)-5-(3,4,5-trimethoxybenzylidene)thiazol-4(5H)-one* (**12**). ^1^H NMR (DMSO-*d*_6_, 500 MHz) δ 12.52 (brs, 1H, NH), 7.95 (d, 1H, *J* = 7.5 Hz), 7.92 (d, 1H, *J* = 8.0 Hz), 7.73 (d, 1H, *J* = 8.5 Hz, 4-H), 7.60 (s, 1H, vinylic H), 7.55–7.46 (m, 3H, 3-H, 6-H, 7-H), 7.16 (d, 1H, *J* = 7.5 Hz, 2-H), 6.75 (s, 2H, 2′-H, 6′-H), 3.66 (s, 6H, 2 × OCH_3_), 3.64 (s, 3H, OCH_3_); ^13^C NMR (DMSO-*d*_6_, 125 MHz) δ 168.2, 153.5, 152.1, 144.4, 139.6, 134.3, 130.4, 129.4, 128.4, 127.8, 127.1, 126.5, 126.4, 125.4, 123.5, 122.7, 116.4, 107.9, 60.6, 56.5; LRMS (ESI−) *m*/*z* 441 (M + Na − 2H)^−^; yield: 72%.

*(Z)-5-(4-Hydroxy-3,5-dimethoxybenzylidene)-2-(naphthalen-1-ylamino)thiazol-4(5H)-one* (**13**). ^1^H NMR (DMSO-*d*_6_, 500 MHz) δ 9.95 (brs, 1H, OH), 7.96 (d, 1H, *J* = 7.5 Hz), 7.91 (d, 1H, *J* = 7.5 Hz), 7.70 (d, 1H, *J* = 8.0 Hz, 4-H), 7.54–7.44 (m, 4H, 3-H, 6-H, 7-H, vinylic H), 7.14 (d, 1H, *J* = 7.0 Hz, 2-H), 6.73 (s, 2H, 2′-H, 6′-H), 3.66 (s, 6H, 2 × OCH_3_); ^13^C NMR (DMSO-*d*_6_, 125 MHz) δ 170.1, 153.8, 148.6, 145.1, 138.6, 134.3, 130.1, 128.4, 128.0, 126.9, 126.4, 126.3, 124.9, 124.4, 123.7, 121.2, 116.4, 108.4, 56.5; LRMS (ESI+) *m*/*z* 429 (M + Na)^+^; LRMS (ESI−) *m*/*z* 405 (M − H)^−^; yield: 82%.

*(Z)-5-(3,5-Di-tert-butyl-4-hydroxybenzylidene)-2-(naphthalen-1-ylamino)thiazol-4(5H)-one* (**14**). ^1^H NMR (CDCl_3_, 500 MHz) δ 9.82 (brs, 1H, NH), 7.95 (d, 1H, *J* = 8.5 Hz), 7.85 (d, 1H, *J* = 8.0 Hz), 7.79 (s, 1H, vinylic H), 7.70 (d, 1H, *J* = 8.0 Hz, 4-H), 7.51 (t, 1H, *J* = 7.5 Hz), 7.44 (t, 1H, *J* = 7.5 Hz), 7.44 (t, 1H, *J* = 7.5 Hz), 7.28 (s, 2H, 2′-H, 6′-H), 7.22 (d, 1H, *J* = 7.5 Hz, 2-H), 5.59 (brs, 1H, OH), 1.39 (s, 18H, 2 × *t*-Bu); ^13^C NMR (DMSO-*d*_6_, 125 MHz) δ 168.5, 156.4, 152.3, 144.7, 139.7, 134.3, 131.5, 128.4, 127.8, 127.5, 127.5, 127.0, 126.4, 126.2, 125.2, 123.5, 119.4, 116.3, 35.0, 30.4; LRMS (ESI+) *m*/*z* 481 (M + Na)^+^; LRMS (ESI−) *m*/*z* 457 (M − H)^−^; yield: 77%.

*(Z)-5-(3-Bromo-4-hydroxybenzylidene)-2-(naphthalen-1-ylamino)thiazol-4(5H)-one* (**15**). ^1^H NMR (DMSO-*d*_6_, 500 MHz) δ 12.19 (brs, 1H, OH), 11.01 (brs, 1H, OH), 7.94 (d, 1H, *J* = 8.0 Hz), 7.90 (d, 1H, *J* = 8.0 Hz), 7.74 (d, 1H, *J* = 8.5 Hz, 4-H), 7.62 (d, 1H, *J* = 2.0 Hz, 2′-H), 7.56–7.48 (m, 3H, 3-H, 6-H, 7-H), 7.53 (s, 1H, vinylic H), 7.24 (dd, 1H, *J* = 8.5, 2.0 Hz, 6′-H), 7.12 (d, 1H, *J* = 7.0 Hz, 2-H), 6.96 (d, 1H, *J* = 8.5 Hz, 5′-H); ^13^C NMR (DMSO-*d*_6_, 125 MHz) δ 168.2, 156.1, 151.6, 144.7, 135.5, 134.3, 130.4, 129.0, 128.4, 127.5, 127.1, 126.6, 126.5, 126.4, 125.3, 123.4, 120.9, 117.3, 116.5, 110.5; LRMS (ESI−) *m*/*z* 423 (M − H)^−^; HRMS (EDA) *m*/*z* C_20_H_14_BrN_2_O_2_S (M + H)^+^ calcd 424.9954, obsd 424.9953, (M + H + 2) calcd 426.9934, obsd 426.9934; yield: 80%.

### 3.2. Mushroom Tyrosinase Activity Assay [42,43]

Test samples (10 μL; compounds **1**–**15** and kojic acid, a positive control) were mixed with a mushroom tyrosinase aqueous solution (20 μL; 500 units/mL) and a substrate solution (170 μL; 345 μM substrate [l-DOPA or l-tyrosine] and 17.2 mM sodium phosphate pH 6.5 buffer) in each well of 96-well microplate. To obtain the IC_50_ values, three to five appropriate concentrations of the test samples were used. After 15 and 30 min of incubation with l-DOPA and l-tyrosine, respectively, the optical density was measured at 475 nm using a VersaMax™ enzyme-linked immunosorbent assay (ELISA) reader (Molecular Devices, Sunnyvale, CA, USA) at 2 min intervals for 30 min to calculate tyrosinase activity. The % tyrosinase inhibition was acquired from the formula (1 − OD_sam_/OD_con_) × 100, where OD_con_ and OD_sam_ represent the optical densities of control and sample, respectively.

### 3.3. Kinetic Experiment Using Mushroom Tyrosinase [38,44,45]

The identification of the tyrosinase inhibition mechanism was carried out by kinetic studies using mushroom tyrosinase. Test samples (10 μL; compounds **10** and **15**) were mixed with a mushroom tyrosinase aqueous solution (20 μL; 100 units/mL) and a substrate solution (170 μL; substrate [l-DOPA] and 17.2 mM sodium phosphate pH 6.5 buffer) in each well of 96-well microplate. To obtain Lineweaver–Burk plots for kinetic analysis, l-DOPA and test samples were used at various concentrations: 0, 1, 2, and 4 μM for **10**; 0, 4, 8, and 16 μM for **15**; and 1, 2, 4, 8, and 16 mM for l-DOPA. During a 30 min incubation period, the initial dopachrome formation rate was calculated by measuring the optical density at 475 nm at 5 min intervals using an ELISA reader. Lineweaver–Burk plots were prepared by plotting the inverse of the substrate concentration against the inverse of the initial dopachrome formation rate in the presence of four different concentrations of compounds **10** and **15**.

### 3.4. In Silico Docking Simulation Using AutoDock Vina [38]

In silico docking simulations of mushroom tyrosinase and compounds **10** and **15** were performed to determine the binding affinity and chemical interactions between tyrosinase and ligands. The 2D structures of ligands (**10** and **15**) drawn using ChemDraw Ultra 12.0 (CambridgeSoft Corporation, Cambridge, MA, USA) were converted to the corresponding 3D structures using Chem3D Pro 12.0 (CambridgeSoft Corporation, Cambridge, MA, USA). The RCSB Protein Data Bank (PDB) was used to obtain the 3D X-ray crystal structure of tyrosinase (*Agaricus bisporus*; PDB ID: 2Y9X). The B–H chains of the X-ray tyrosinase protein structure were deleted, and the remaining A chain was docked with each 3D ligand structure using AutoDock Vina 1.1.3 (Scripps Research, La Jolla, San Diego, CA, USA). The position where tropolone, the original ligand of 2Y9X, was bound, was used as the active site. Plausible chemical interactions between the ligands and enzymes were obtained using LigandScout 4.4 (Cambridge, UK).

### 3.5. Cell Culture

B16F10 cells were cultured in a solution containing 1% penicillin–streptomycin solution (100×), 10% heat-inactivated fetal bovine serum (FBS), and Dulbecco’s modified Eagle’s medium (DMEM) (Welgene, Gyeongsan-si, Republic of Korea) at 37 °C in a humidified environment containing 5% CO_2_.

### 3.6. Cell Viability on B16F10 Cells [46]

The viability of B16F10 cells in the presence of compounds **10** and **15** was assayed. A 96-well microplate containing 1 × 10^3^ B16F10 cells per well was incubated at 37 °C in a humid environment containing 5% CO_2_ for 24 h. Test samples (**10** and **15**) were added to each well at concentrations of 0, 1, 2, 5, 10, and 20 μM and incubated for 72 h. Each well was exposed to a 10 μL EZ-Cytox solution (EZ-500^®^, DoGenBio, Seoul, Republic of Korea) and incubated for 2 h. To determine cell viability, well optical density was measured at 450 nm using an ELISA reader.

### 3.7. Melanin Content Assay and Cellular Tyrosinase Activity Assay in B16F10 Cells [47]

A 6-well microplate containing 5 × 10^3^ B16F10 cells per well was incubated at 37 °C for 24 h. Compound **10** was added to each well at 5, 10, and 20 μM and incubated for 1 h, and then stimulators (1 μM α-MSH and 200 μM IBMX) were added to each well before 72 h incubation. Kojic acid (20 μM) was used for comparing biological activity.

For the melanin content quantification, the cells were lysed at 60 °C for 1 h in 100 μL 1N-NaOH solution containing 10% DMSO after washing 2 times with phosphate-buffered saline (PBS). The cell lysates were transferred to a 96-well microplate and the optical density was measured at 405 nm using an ELISA reader to calculate the melanin content in each well.

To measure the cellular tyrosinase activity, the cells incubated for 72 h were washed 2 times with PBS, exposed to a 100 μL lysis buffer solution containing 5 μL 2 mM phenylmethylsulfonyl fluoride, 5 μL 20% Triton X-100, and 90 μL 50 mM phosphate buffer (pH 6.5), and cultivated for 30 min at −80 °C. After the lysates were centrifuged at 4 °C for 10 min at 10,000× *g*, an aliquot (80 μL) of the supernatants was added to each well of a 96-well microplate containing 20 μL of 10 mM l-DOPA. Data for the optical density at 475 nm were obtained every 1 min for 10 min at 37 °C using an ELISA reader to determine cellular tyrosinase activity.

### 3.8. In Situ Cellular Tyrosinase Activity Assay [48,49]

A 24-well plate containing 1 × 10^3^ B16F10 cells per well was incubated at 37 °C for 24 h. The cells were pre-treated with kojic acid (20 μM) used for comparing tyrosinase activity or compound **10** (20, 10, and 5 μM) for 1 h and exposed to stimulators (200 μM IBMX and 1 μM α-MSH). After incubation for 72 h, the cells were fixed for 50 min using paraformaldehyde (4%), rinsed twice with PBS, and permeabilized for 2 min using Triton X-100 (0.1%). The cells were washed with PBS and 500 μL of 2 mM l-DOPA solution was added. After the cells were incubated at 37 °C for 2 h, stained images were obtained using a camera connected to a microscope (Motic, Hong Kong, China).

### 3.9. Zebrafish Depigmentation Assay Using Zebrafish Embryos [50,51,52]

To obtain zebrafish (wild type, *Danio rerio*) eggs for the experiment, six male and nine female zebrafish were raised separately in oxygen-connected tanks. Dried brine shrimp (*Artemia salina*, San Francisco Bay Brand, San Francisco, CA, USA) were fed as food three times per day. They were raised at 28 °C with 10 h of light blocking and 14 h of light exposure. The evening before obtaining zebrafish eggs, three females and two males were placed together in three zebrafish mating cages, and the light was blocked. The following morning, mating was induced by exposure to light. Harvested zebrafish eggs, obtained free of charge from the Zebrafish Center for Disease Modeling (Chungnam National University, Daejeon, Republic of Korea), were kept in an E3 embryonic solution containing methylene blue (MB) until the dechorionation process was performed. To ensure that test samples could be efficiently delivered to zebrafish embryos, the chorion of the zebrafish embryos was removed using PRONASE^®^ (Merck Millipore, Burlington, MA, USA) at 24 hpf. Five dechorionated embryos were dispensed into each well of a 48-well plate containing the E3 embryonic solution (500 μL) without MB. Test samples (**15** [0.1 and 0.3 mM] and kojic acid [20 mM], a positive control) were added to each well at 28 hpf and cultured in an incubator set at 28 °C for 48 h. To determine the depigmentation effect of the test samples, zebrafish larvae were anesthetized with tricaine methanesulfonate, placed on 1% methylcellulose blocks, and photographed using a SMZ745T Nikon stereoscopic microscope (Tokyo, Japan) at 76 hpf. Dorsal and lateral views of zebrafish larvae were photographed separately. The pigmented areas in the zebrafish larval photographs were determined using an ATTO CS analyzer 3.2 (Tokyo, Japan).

To quantify the tyrosinase activity in zebrafish larvae, 60 zebrafish larvae at 76 hpf were divided into three Eppendorf tubes and centrifuged at 10,000× *g* for 10 min at 4 °C. After removing the supernatant, the residue was lysed using radioimmunoprecipitation assay buffer (100 μL) (Biosedang, Gyeonggi-do, Republic of Korea) and homogenized for 5 min using a sonicator. The lysate was centrifuged at 10,000× *g* for 10 min at 4 °C. The supernatants were transferred to a new Eppendorf tube and the protein content was quantified using a bicinchoninic acid protein assay kit (Thermo Fisher Scientific, Waltham, MA, USA). An aliquot (100 μL) of 1 mM l-DOPA was added to the new Eppendorf tube containing a total of 300 μg of protein in 100 μL of protein lysate. Eppendorf tubes were incubated for 1 h at 37 °C and then transferred to each well of a 96-well plate. The absorbance of each well was measured at 475 nm to determine the tyrosinase activity of zebrafish larvae.

### 3.10. Assay for DPPH Radical Scavenging [53]

DPPH was dissolved in methanol to prepare a 0.2 mM solution, and test samples (vitamin C [l-ascorbic acid, LA; positive control] and **1**–**15**) were dissolved in DMSO to prepare a 5 mM solution. The DPPH methanolic solution (180 μL) was added to the test sample DMSO solution (20 μL) in each well of a 96-well microplate and kept in a place without light for 30 min at 22 °C. Using a VersaMax™ ELISA reader, the optical density of each well was recorded at 517 nm to determine DPPH radical-scavenging activity.

### 3.11. Assay for ABTS^+^ Radical Scavenging [54]

To acquire ABTS^+^ radicals, ABTS (14 mM in 20 mL H_2_O) was mixed with K_2_S_2_O_8_ (4.9 mM in 20 mL H_2_O), an oxidant agent, and stored at 22 °C in a place without light for 15 h. The absorbance of the prepared ABTS^+^ radical solution was adjusted to 0.70 ± 0.01 at 732 nm by adding methanol. Test samples (Trolox [positive control] and **1**–**15**) were dissolved in a co-solvent (DMSO:EtOH = 1:10 [*v*/*v*]). The diluted ABTS^+^ radical solution (90 μL) was added to the test sample solution (10 μL) in each well of a 96-well plate and held at 22 °C in the dark for 2 min. The optical density of each well was recorded at 732 nm every 1 min for 10 min using an ELISA reader. The final concentration of test samples was 100 μM. %Radical-scavenging activity was determined using the formula 100 × [(Abs_con_ − Abs_sam_)/Abs_con_], where Abs_con_ and Abs_sam_ represent the optical densities of the control and the samples, respectively.

### 3.12. Assay for ROS Scavenging

To obtain 2′,7′-dichlorodihydrofluorescein (DCFH) solution, esterase (3 units/2 mL) was mixed with 2.5 mM DCFH diacetate and 50 mM phosphate buffer (pH 7.4). After incubating the mixture at 22 °C for 25 min, it was kept in the dark. 3-Morpholinosydnonimine (SIN-1) and compounds **1**–**15** were dissolved in phosphate buffer and DMSO, respectively. SIN-1 (10 µL) and compound (10 µL) were added to each well of a 96-well black plate containing phosphate buffer (180 µL), and the concentrations of SIN-1 and compound were 10 and 40 μM, respectively. Trolox (40 μM) was used as a positive material. After the black plate was kept for 10 min in a dark place, the DCFH solution (50 µL) was added to each well of the black plate. To measure the fluorescence of 2′,7′-dichlorofluorescein at 535 nm, a microplate reader (Berthold Advances GmbH &Co., Bad Wildbad, Germany) was utilized with an excitation wavelength of 485 nm. All experiments were independently conducted three times.

### 3.13. Statistical Analysis

All experiments were independently conducted thrice to achieve statistical significance. Data are expressed as the mean ± SEM. Significance was determined by one-way ANOVA followed by a Newman–Keuls test. Significance was set at *p* < 0.05.

## 4. Conclusions

Based on the hybrid structure of 2-aminothiazol-4(5*H*)-one and β-phenyl-α,β-unsaturated carbonyl, compounds **1**–**15** were designed and synthesized as potential tyrosinase inhibitors. Compounds **10** and **15** were stronger mushroom tyrosinase inhibitors than kojic acid, and kinetic experiments demonstrated that these compounds competitively inhibited tyrosinase. In silico docking simulation showed that compounds **10** and **15** bind to the active site of tyrosinase with high binding affinities of −8.4 and −8.9 kcal/mol, respectively, supporting the kinetic results. In B16F10 cell-based experiments, **10** strongly inhibited the cellular tyrosinase activity and melanin production. In depigmentation experiments using zebrafish embryos, compound **15** reduced the pigmentation of zebrafish larvae to a similar extent as kojic acid at concentrations hundreds of times lower than kojic acid.

## Data Availability

The data presented in this study are available in the article and Appendix A.

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
