# Peer review of "Design, Synthesis, and Antioxidant and Anti-Tyrosinase Activities of (Z)-5-Benzylidene-2-(naphthalen-1-ylamino)thiazol-4(5H)-one Analogs: In Vitro and In Vivo Insights"

_molecules, 2025, doi:10.3390/molecules30020289_

Round 1
Reviewer 1 Report
Comments and Suggestions for Authors
The manuscript by Hyung Ryong Moon et al. entitled “Design, synthesis, and antioxidant and anti-tyrosinase activities of (Z)-5-benzylidene-2-(naphthalen-1-ylamino)thiazol-4(5H)-one analogs: In vitro and in vivo insights” reports the synthesis of fifteen compounds constructed on the hybrid structure combining a β-phenyl-α,β-unsaturated carbonyl template and a 2-aminothiazol-4(5H)-one scaffold as potential novel anti-tyrosinase substances.
In the reviewed manuscript the syntheses and activity studies were planned correctly. However, the conclusions and methodology require improvement and are inconsistent.
After careful evaluation of the manuscript I have some comments:
Synthesis
1. The structures of the obtained target compounds were confirmed by 1H and 13C (NMR) and LRMS, but there is no confirmation of purity. Nevertheless Authors had written in Materials and methods:
- line 398-9: The purity of 16 was >98%.
- line 406-7: The purity of 17 was >96%.
- Line 418: with a purity of 98% or more.
2. Authors should include either elemental analysis or a quantitative HPLC/GC trace to support bulk purity. Please note that LRMS is not proof of purity, and NMR silent impurities will not show up in NMR studies.
3. I agree with the authors that “The alkenes of the target compounds had (Z)-geometries.” and with their arguments, but in 13C NMR spectra are present multiplicity, which may suggest isomerization or instability. In this regard, quantitative HPLC would be helpful.
4. In Supporting Information is lacking the MS spectra for compound 4, must be completed. And for compound 16 there is only 1H NMR, but this may be because of the intermediate compound.
Activity
5. In the literature, we can find tyrosinase inhibitory activity (IC50) for kojic acid in range 15-20 mM for diphenolase activity and 70-90 mM for monophenolase activity (mushroom tyrosinase), but in presented studies (Table 1.) this value is 27.41 mM and 24.09 mM, respectively. Please explain it.
6. How do the Authors explain the difference in the activity of the compound 10 (2,4-dihydroxy) relative to the activity of the compound 8 (3,4-dihydroxy)? Where does such a significant difference come from - the most active compound 10 to the practically inactive one 8? Especially since the DOPA substrate has analogous substituents in the aromatic ring.
For derivatives 3 and 4 (3,4- and 2,4-dimethoxy) or 9 and 11 (3,4- and 2,-difluoro) we do not observe such significant changes.
In addition, compound 8 is the strongest antioxidant compound in all tested methods.
7. Unfortunately docking simulation were prepared only for compound 10 and 15. To be able to draw correct conclusions about SAR, a Molecular docking study should be performed for all compounds. Please complete it.
8. For the antioxidant ability of obtained compounds I consider it unnecessary to emphasize the average activity of compound 10 in relation to other more active compounds, e.g. compound 8.
9. Why for in vivo depigmentation effect in zebrafish larvae experiments compound 15 was chosen?
Previous cytotoxicity on B16F10 cells studies excluded compound 15 from further experiments?
The depigmentation activity tests should be repeated for compound 10.
10. For the treatment of skin hyperpigmentation disorders it is necessary to conducted studies with human cell lines, e.g. melanoma cell lines.
11. In Conslusion phrase: “Contrastingly, in depigmentation experiments using zebrafish embryos, compound 15 reduced the depigmentation of zebrafish larvae” is misleading. Other compounds using zebrafish larvae have not been studied.
12.The 52 cited references are mainly from the last 20 years (with exceptions: reference 41. 1934 or reference 42. 1953) and it doesn’t include an excessive number of self-citations.
13. The manuscript is a relatively well-written, but there are small misprints:
- e.g. line 370 - (Figure 9C) instead of (Figure 9B),
- Reference 27. Should be completed, 2015.
Author Response
We would like to thank the reviewer for kind and detailed comments and suggestions that helped improve the quality of our manuscript.
Reviewer 1.
The manuscript by Hyung Ryong Moon et al. entitled “Design, synthesis, and antioxidant and anti-tyrosinase activities of (Z)-5-benzylidene-2-(naphthalen-1-ylamino)thiazol-4(5H)-one analogs: In vitro and in vivo insights” reports the synthesis of fifteen compounds constructed on the hybrid structure combining a β-phenyl-α,β-unsaturated carbonyl template and a 2-aminothiazol-4(5H)-one scaffold as potential novel anti-tyrosinase substances.
In the reviewed manuscript the syntheses and activity studies were planned correctly. However, the conclusions and methodology require improvement and are inconsistent.
After careful evaluation of the manuscript I have some comments:
Synthesis
1. The structures of the obtained target compounds were confirmed by 1H and 13C (NMR) and LRMS, but there is no confirmation of purity. Nevertheless Authors had written in Materials and methods:
- line 398-9: The purity of 16 was >98%.
- line 406-7: The purity of 17 was >96%.
- Line 418: with a purity of 98% or more.
Response:
Thank you for your detailed comment. We had measured the purity of the compound based on the 1H NMR spectra. We deleted the contents related to the purity.
2. Authors should include either elemental analysis or a quantitative HPLC/GC trace to support bulk purity. Please note that LRMS is not proof of purity, and NMR silent impurities will not show up in NMR studies.
Response:
Thank you for your kind comment. Instead of elemental analysis, we have added high-resolution mass data for compounds (8 and 10) used in cell and in vivo experiments in section 3.1.1., and the high-resolution mass spectra of these compounds have been added to Supplementary Information. In addition, we have added high-resolution mass data for compound 4 lacking the LR mass spectra.
3. I agree with the authors that “The alkenes of the target compounds had (Z)-geometries.” and with their arguments, but in 13C NMR spectra are present multiplicity, which may suggest isomerization or instability. In this regard, quantitative HPLC would be helpful.
Response:
Thank you for your kind comment. The multiplicity shown in 13C NMR spectra is due to fluorine and not to isomerization or instability. Because a fluorine atom has the spin quantum number I = 1/2, heteronuclear 13C-19F coupling is observed when there are fluorine atoms in an organic compound. As compounds 9 and 11 have two fluorine atoms, their 13C NMR spectra show multiplicity due to 13C-19F coupling.
4. In Supporting Information is lacking the MS spectra for compound 4, must be completed. And for compound 16there is only 1H NMR, but this may be because of the intermediate compound.
Response:
Thank you for your detailed and kind comment. We have added high-resolution mass data for compound 4. (Please see section 3.1.1. and Supplementary Information)
Activity
5. In the literature, we can find tyrosinase inhibitory activity (IC50) for kojic acid in range 15-20 mM for diphenolase activity and 70-90 mM for monophenolase activity (mushroom tyrosinase), but in presented studies (Table 1.) this value is 27.41 mM and 24.09 mM, respectively. Please explain it.
Response:
Thank you for your valuable comment.
The following references report that kojic acid has IC50 values ​​of 10-30 μM for l-tyrosine and 10-50 μM for l-dopa. Various different IC50 values ​​are reported in the literature, which is thought to be due to the differences in the activity and amount of tyrosinase used in the experiment and the subtle differences in the experimental method.
1. Reviews in Agricultural Science (Review), 2019, 7, 41-58. l-tyrosine (IC50 = 13.2 μM), l-dopa (IC50 = 26.5 μM).
2. ACS omega, 2018, 3, 17236-17245. l-tyrosine (IC50 = 9.78 μM), l-dopa (IC50 = 9.21 μM)
3. Scientific Reports, 2016, 6, 34993. l-tyrosine (IC50 = 26.8 μM), l-dopa (IC50 = 52 μM)
4. Journal of enzyme inhibition medicinal chemistry, 2016, 31(5), 742-747. l-tyrosine (IC50 = 11.5 μM), l-dopa (IC50 = 24.1 μM)
6. How do the Authors explain the difference in the activity of the compound 10(2,4-dihydroxy) relative to the activity of the compound 8(3,4-dihydroxy)? Where does such a significant difference come from - the most active compound 10 to the practically inactive one 8? Especially since the DOPA substrate has analogous substituents in the aromatic ring.
For derivatives 3 and 4 (3,4- and 2,4-dimethoxy) or 9 and 11 (3,4- and 2,-difluoro) we do not observe such significant changes.
In addition, compound 8 is the strongest antioxidant compound in all tested methods.
Response:
Thank you for your valuable comment.
We believe that compound 8 has little effect on tyrosinase inhibition because it acts as a substrate rather than an inhibitor of tyrosinase. This is because the substituents (3,4-dihydroxy) on the aromatic ring of compound 8 are identical to those of the DOPA substrate.
On the other hand, compound 10 binds strongly to tyrosinase as shown in the docking simulation results, and it also has hydroxyl groups at positions 2 and 4 of the aromatic ring, which makes hydroxylation by tyrosinase at position 3 difficult due to steric hindrance of 2,4-dihydroxyl, so it seems to act as a strong tyrosinase inhibitor.
Unlike compound 8, derivatives 3 and 9, which have 3,4-dimethoxy and 3,4-difluoro, respectively, cannot act as tyrosinase substrates. Derivatives 3 and 4 (3,4- and 2,4-dimethoxy) or 9 and 11 (3,4- and 2,4-difluoro) have similar binding affinities to tyrosinase, so that no significant change is observed for these derivatives as observed between compounds 8 and 10.
7. Unfortunately docking simulation were prepared only for compound 10and 15. To be able to draw correct conclusions about SAR, a Molecular docking study should be performed for all compounds. Please complete it.
Response:
Thank you for your valuable comment.
We performed docking simulation of all compounds with tyrosinase, and the results are included in the Supplementary Information.
8. For the antioxidant ability of obtained compounds I consider it unnecessary to emphasize the average activity of compound 10in relation to other more active compounds, e.g. compound 8.
Response:
Thank you for your valuable comment.
As suggested by the reviewer, the antioxidant activity of compound 10 was de-emphasized.
9. Why for in vivo depigmentation effect in zebrafish larvae experiments compound 15 was chosen?
Previous cytotoxicity on B16F10 cells studies excluded compound 15 from further experiments?
The depigmentation activity tests should be repeated for compound 10.
Response:
Thank you for your valuable comment.
Compound 15 showed cytotoxicity in B16F10 cell-based experiments, but not in zebrafish larvae experiments. In contrast, compound 10 showed no cytotoxicity in B16F10 cell-based experiments, but compound 10 appeared to be toxic to zebrafish larvae. In preliminary experiments using zebrafish embryos, we observed that the zebrafish larvae treated with compound 10 showed a slightly warped shape, which may be related to the toxicity to zebrafish larvae. Therefore, compound 10 was excluded from the main in vivo experiments.
10. For the treatment of skin hyperpigmentation disorders it is necessary to conducted studies with human cell lines, e.g. melanoma cell lines.
Response:
Thank you for your valuable comment.
Previously, we purchased A375P cells (human melanoma cell line) and seeded them, but proliferation was almost nonexistent, so we were unable to culture them. We plan to find conditions to optimize culture conditions for A375P cells and conduct experiments using this human melanoma cell line in the future.
It takes a considerable amount of time to optimize the conditions for culturing A375P cells. We ask for your understanding so that we can conduct this experiment in the future.
11. In Conslusion phrase: “Contrastingly, in depigmentation experiments using zebrafish embryos, compound 15 reduced the depigmentation of zebrafish larvae” is misleading. Other compounds using zebrafish larvae have not been studied.
Response:
Thank you for your detailed comment.
We have revised our mistake as follows: In depigmentation experiments using zebrafish embryos, compound 15 reduced the pigmentation of zebrafish larvae to a similar extent as kojic acid at concentrations hundreds of times lower than kojic acid.
After preliminary experiments with all compounds using zebrafish larvae, we selected compound 15 as the most promising compound for the main experiment. Since the other compounds showed lower depigmentation efficacy than 15 in the preliminary experiments, these compounds were excluded from the main experiment.
12. The 52 cited references are mainly from the last 20 years (with exceptions: reference 41. 1934 or reference 42. 1953) and it doesn’t include an excessive number of self-citations.
Response:
Thank you for your kind comment.
13. The manuscript is a relatively well-written, but there are small misprints:
- e.g. line 370 - (Figure 9C) instead of (Figure 9B),
- Reference 27. Should be completed, 2015.
Response:
Thank you for your detailed comment.
We have revised “Figure 9B” to “Figure 9C”, and have completed Reference 27.
Reviewer 2 Report
Comments and Suggestions for Authors
The article under the title “Design, synthesis, and antioxidant and anti-tyrosinase activities of (Z)-5-benzylidene-2-(naphthalen-1-ylamino)thiazol-4(5H)-one analogs: In vitro and in vivo insights” by Jung and coworkers presents results on the synthesis, structural, and biological examination of the novel derivatives of (Z)-5-benzylidene-2-(naphthalen-1-ylamino)thiazol-4(5H)-one. The article is well-written and the obtained experimental and theoretical results are nicely explanined. The article could be of potential interest to the readers of the Molecules, although there are some points that should be addressed before the final decision. Therefore, my recommendation is MAJOR REVISION.
The authors should answer the following:
1. In vitro and in vivo should be written in italic
2. “Melanin has a positive viewpoint of protecting” – rephrase
3. The authors should give an overview of the substituents that were used for the preparation of compounds 1-15 in the first paragraph of section 2.1
4. The authors should compare the results of SAR analysis with data from literature, especially bearing in mind present substituents
5. A plausible explanation of the effects of these substituents should be given in the main text.
6. The authors should outline the structural parameters of two compounds that are important for the binding affinity to selected protein.
7. The differences and similarities in structures of kojic acid and mentioned derivatives 10 i 15 should be given, as they might be important for the interactions with amino acids
8. The authors should compare biological activity of these compounds to the literature values for similar compounds.
9. The authors should outline the structural parameters that are important for the antioxidant activity and some of the limitations for the used antioxidant activity tests.
Author Response
We would like to thank the reviewer for kind and detailed comments and suggestions that helped improve the quality of our manuscript.
Reviewer 2.
The article under the title “Design, synthesis, and antioxidant and anti-tyrosinase activities of (Z)-5-benzylidene-2-(naphthalen-1-ylamino)thiazol-4(5H)-one analogs: In vitro and in vivo insights” by Jung and coworkers presents results on the synthesis, structural, and biological examination of the novel derivatives of (Z)-5-benzylidene-2-(naphthalen-1-ylamino)thiazol-4(5H)-one. The article is well-written and the obtained experimental and theoretical results are nicely explanined. The article could be of potential interest to the readers of the Molecules, although there are some points that should be addressed before the final decision. Therefore, my recommendation is MAJOR REVISION.
The authors should answer the following:
1. In vitro and in vivo should be written in italic
Response:
Thank you for your detailed comment.
As suggested by the reviewer, the words ‘in vitro’ and ‘in vivo’ were italicized.
2. “Melanin has a positive viewpoint of protecting” – rephrase
Response:
Thank you for your detailed comment.
The phrase was revised as follows: Melanin plays a role in protecting the skin.
3. The authors should give an overview of the substituents that were used for the preparation of compounds 1-15 in the first paragraph of section 2.1
Response:
Thank you for your detailed comment.
As per the reviewer, an overview of the substituents on the phenyl ring of the final compounds 1–15 were added in the first paragraph of section 2.1.
〔The substituents on the final compounds 1–15 included hydroxy, methoxy, ethoxy, fluorine, bromine and tert-butyl groups, which either donate (hydroxyl and alkoxy) or withdraw (fluorine and bromine) electrons to the phenyl ring or provide bulkiness (tert-butyl)〕.
4. The authors should compare the results of SAR analysis with data from literature, especially bearing in mind present substituents
Response:
Thank you for your valuable comment. As suggested by the reviewer, we compared the results of SAR analysis with data from literature.
〔We compared the SAR results with (E)-benzylidene-1-indanone (BID) derivatives [37]. Just as the addition of a 2-hydroxy substituent to the BID compound having a 4-hydroxy substituent strongly increased the tyrosinase inhibitory activity (IC50 value: 39.7 μM vs. 0.03 μM), compound 10, which was obtained by adding a 2-hydroxy substituent to compound 2 having a 4-hydroxy substituent, strongly increased the tyrosinase inhibitory activity (IC50 value: 148.1 μM vs. 2.9 μM). When bromine was added to the BID compound containing a 4-hydroxy substituent, the tyrosinase inhibitory activity decreased (IC50 value: 39.7 μM vs. >200 μM), but when bromine was added to compound 2 having a 4-hydroxy substituent, the inhibitory activity increased (IC50 value: 148.1 μM vs. 6.9 μM). As in many literatures [22,24,38], BID compound having a 4-hydroxy substituent showed stronger tyrosinase inhibitory activity than BID compound having a 4-methoxy substituent (IC50 value: 39.7 μM vs. >200 μM), but compound 2 having a 4-hydroxy substituent showed weaker tyrosinase inhibitory activity than compound 1 having a 4-methoxy substituent (IC50 value: 148.1 μM vs. 58.4 μM).〕
5. A plausible explanation of the effects of these substituents should be given in the main text.
Response:
Thank you for your valuable comment.
As per the reviewer, the effects of these substituents were added in section 2.1.
〔The substituents on the final compounds 1–15 included hydroxy, methoxy, ethoxy, fluorine, bromine and tert-butyl groups, which either donate (hydroxyl and alkoxy) or withdraw (fluorine and bromine) electrons to the phenyl ring or provide bulkiness (tert-butyl)〕.
6. The authors should outline the structural parameters of two compounds that are important for the binding affinity to selected protein.
Response:
Thank you for your valuable comment.
As per the reviewer, an outline of the structural parameters of two compounds that are important for binding affinity to mushroom tyrosinase was added in section 2.4.
〔Compounds 10 and 15 hydrophobically interacted with the same three amino acid residues (Phe264, Val283, and Ala286) in their naphthalene rings. In addition, compounds 10 and 15 increased binding affinity by allowing the hydroxyl group at position 2 and position 4 of the phenyl ring to participate in hydrogen bonding, respectively.〕
7. The differences and similarities in structures of kojic acid and mentioned derivatives 10 i 15 should be given, as they might be important for the interactions with amino acids
Response:
Thank you for your valuable comment.
As per the reviewer, the differences in structures of kojic acid and mentioned derivatives 10 and 15 were added in section 2.4.
〔Kojic acid is known to inhibit the activity of tyrosinase by chelating the copper of tyrosinase using the α-hydroxy ketone. Compounds 10 and 15 do not have a chemical structure that can chelate copper. These compounds were designed to bind well to the surrounding amino acids at the active site of tyrosinase. Therefore, as shown in the docking simulation results, kojic acid and these compounds had very different interaction patterns with tyrosinase amino acid residues.〕
8. The authors should compare biological activity of these compounds to the literature values for similar compounds.
Response:
Thank you for your valuable comment.
The answer is attached together with the answer to item 4.
9. The authors should outline the structural parameters that are important for the antioxidant activity and some of the limitations for the used antioxidant activity tests.
Response:
Thank you for your valuable comment.
〔To summarize the structures important for antioxidant activity, in general, the greater the number of hydroxyl groups on the phenyl ring, the stronger the antioxidant activity was exhibited, and in particular, the catechol (3,4-dihydroxyphenyl) structure showed strong antioxidant activity in various antioxidant experiments.〕
Round 2
Reviewer 2 Report
Comments and Suggestions for Authors
The authors have answered all of the questions properly. The manuscript is suitable for publication.